# Manganese is a physiologically relevant TORC1 activator in yeast and mammals

Raffaele Nicastro[1†], Hélène Gaillard[2,3†], Laura Zarzuela[2], Marie-Pierre Péli-Gulli[1], Elisabet Fernández-García[2,3], Mercedes Tomé[2], Néstor García-Rodríguez[2,3], Raúl V Durán[2], Claudio De Virgilio[1*], Ralf Erik Wellinger[2,3*]

[1]University of Fribourg, Department of Biology, Fribourg, Switzerland; [2]Centro Andaluz de Biología Molecular y Medicina Regenerativa - CABIMER, Consejo Superior de Investigaciones Científicas, Universidad de Sevilla, Seville, Spain; [3]Departamento de Genética, Facultad de Biología, Universidad de Sevilla, Seville, Spain

**\*For correspondence:**
Claudio.DeVirgilio@unifr.ch
(CDV);
wellinger@us.es (REW)

[†]These authors contributed equally to this work

**Competing interest:** The authors declare that no competing interests exist.

**Abstract** The essential biometal manganese (Mn) serves as a cofactor for several enzymes that are crucial for the prevention of human diseases. Whether intracellular Mn levels may be sensed and modulate intracellular signaling events has so far remained largely unexplored. The highly conserved target of rapamycin complex 1 (TORC1, mTORC1 in mammals) protein kinase requires divalent metal cofactors such as magnesium ($Mg^{2+}$) to phosphorylate effectors as part of a homeostatic process that coordinates cell growth and metabolism with nutrient and/or growth factor availability. Here, our genetic approaches reveal that TORC1 activity is stimulated in vivo by elevated cytoplasmic Mn levels, which can be induced by loss of the Golgi-resident $Mn^{2+}$ transporter Pmr1 and which depend on the natural resistance-associated macrophage protein (NRAMP) metal ion transporters Smf1 and Smf2. Accordingly, genetic interventions that increase cytoplasmic $Mn^{2+}$ levels antagonize the effects of rapamycin in triggering autophagy, mitophagy, and Rtg1-Rtg3-dependent mitochondrion-to-nucleus retrograde signaling. Surprisingly, our in vitro protein kinase assays uncovered that $Mn^{2+}$ activates TORC1 substantially better than $Mg^{2+}$, which is primarily due to its ability to lower the $K_m$ for ATP, thereby allowing more efficient ATP coordination in the catalytic cleft of TORC1. These findings, therefore, provide both a mechanism to explain our genetic observations in yeast and a rationale for how fluctuations in trace amounts of Mn can become physiologically relevant. Supporting this notion, TORC1 is also wired to feedback control mechanisms that impinge on Smf1 and Smf2. Finally, we also show that $Mn^{2+}$-mediated control of TORC1 is evolutionarily conserved in mammals, which may prove relevant for our understanding of the role of Mn in human diseases.

## Editor's evaluation

Your manuscript uses budding yeast to uncover a new input for the central metabolic regulator TOR complex 1 (TORC1) – manganese (Mn) levels and also demonstHelene rates that this dependence on Mn is conserved in humans. The combination of both in vivo and in vitro approaches as well as the demonstration of conservation of this phenomenon make the manuscript both broad and deep. TORC1 is already clearly a central coordinator of multiple inputs to guide cellular decisions of catabolism vs anabolism. Information on an additional way to modulate its activity is highly influential on both basic cell biology as well as therapeutic research.

## Introduction

Manganese (Mn) is a vital trace element that is required for the normal activity of the brain and nervous system by acting, among other mechanisms, as an essential, divalent metal cofactor for enzymes such as the mitochondrial enzyme superoxide dismutase 2 (*Weisiger and Fridovich, 1973*), the apical activator of the DNA damage response serine/threonine kinase ATM (*Chan et al., 2000*) or the $Mn^{2+}$-activated glutamine synthetase (*Wedler and Denman, 1984*). However, $Mn^{2+}$ becomes toxic when enriched in the human body (*Couper, 1837*). While mitochondria have been proposed as a preferential organelle where $Mn^{2+}$ accumulates and unfolds its toxicity by increasing oxidative stress and thus mitochondrial dysfunction (*Aguirre and Culotta, 2012*), the molecular mechanisms of $Mn^{2+}$ toxicity in humans are also related to protein misfolding, endoplasmic reticulum (ER) stress, and apoptosis (*Harischandra et al., 2019*). $Mn^{2+}$ homeostasis is coordinated by a complex interplay between various metal transporters for $Mn^{2+}$ uptake and intracellular $Mn^{2+}$ distribution and represents an essential task of eukaryotic cells, which is also of vital importance specifically for neuronal cell health (*Horning et al., 2015*).

Much of our knowledge on $Mn^{2+}$ transport across the plasma membrane into the ER, the Golgi, endosomes, and vacuoles comes from studies in *Saccharomyces cerevisiae* (outlined in *Figure 1A*). Typically, $Mn^{2+}$ is shuttled across membranes by transporters that belong to the natural resistance-associated macrophage protein (NRAMP) family, which are highly conserved metal transporters responsible for iron (Fe) and $Mn^{2+}$ uptake (*Supek et al., 1996*). Not surprisingly, therefore, NRAMP orthologs have been found to cross-complement functions in yeast, mice, and humans (*Sacher et al., 2000*). One of the best-studied NRAMPs is the yeast plasma membrane protein Smf1. Interestingly, extracellular Fe or $Mn^{2+}$ supplementation triggers Bsd2 adaptor protein-dependent, Rsp5-mediated ubiquitination of Smf1, which initiates its sorting through the endocytic multivesicular body pathway and subsequent lysosomal degradation (*Eguez et al., 2004*; *Liu and Culotta, 1999*). The Smf1 paralogs Smf2 and Smf3 are less well studied, but Smf2 is predominantly localized at endosomes and its levels decrease under conditions of Mn or Fe overload (*Liu et al., 1997*; *Moreno-Cermeño et al., 2010*). Within cells, the P-type ATPase Pmr1 (also known as Bsd1) represents a key transporter that shuttles $Ca^{2+}$ and $Mn^{2+}$ ions into the Golgi lumen. Its loss leads to increased levels of $Mn^{2+}$ in the cytoplasm due to defective detoxification (*Lapinskas et al., 1995*). Noteworthy, several phenotypes associated with loss of Pmr1 have been shown to arise as a consequence of $Mn^{2+}$ accumulation in the cytoplasm, including telomere shortening, genome instability, and bypass of the superoxide dismutase Sod1 requirement (*Bolton et al., 2002*; *García-Rodríguez et al., 2012*; *Lapinskas et al., 1995*; *Lue et al., 2005*).

TORC1/mTORC1 is a central, highly conserved controller of cell growth and aging in eukaryotes. It coordinates the cellular response to multiple inputs, including nutritional availability, bioenergetic status, oxygen levels, and, in multicellular organisms, the presence of growth factors (*Albert and Hall, 2015*; *Laplante and Sabatini, 2012*). In response to these diverse cues, TORC1 regulates cell growth and proliferation, metabolism, protein synthesis, autophagy, and DNA damage responses (*Liu and Sabatini, 2020*; *Sancak et al., 2008*). In *S. cerevisiae*, which played a pivotal role in the discovery and dissection of the TOR signaling network (*Heitman et al., 1991*), TORC1 is mainly localized on the surfaces of vacuoles and endosomes (*Betz and Hall, 2013*; *Hatakeyama et al., 2019*) where it integrates, among other cues, amino acid signals through the Rag GTPases and Pib2 (*Nicastro et al., 2017*; *Tanigawa et al., 2021*). In mammals, amino acids also activate the Rag GTPases, which then recruit mTORC1 to the lysosomal surface where it can be allosterically activated by the small GTPase Rheb (Ras homolog expressed in brain) that mediates the presence of growth factors and sufficient energy levels (*Binda et al., 2009*; *Sancak et al., 2008*; *Sancak et al., 2010*; *Wedaman et al., 2003*). Interestingly, mTORC1 regulates cellular Fe homeostasis (*Bayeva et al., 2012*), but how it may be able to sense Fe levels remains largely unknown. In addition, although TORC1/mTORC1 requires divalent metal ions to coordinate ATP at its catalytic cleft (*Brunn et al., 1996*; *Withers et al., 1997*), it is currently not known whether these or any other trace elements may play a physiological or regulatory role in controlling its activity.

In yeast, genetic evidence links high levels of cytoplasmic $Mn^{2+}$ to TORC1 function, as loss of Pmr1 confers rapamycin resistance (*Devasahayam et al., 2007*). However, the underlying molecular mechanism(s) by which $Mn^{2+}$ may mediate rapamycin resistance remains to be explored. Here, we show that $Mn^{2+}$ uptake by NRAMP transporters modulates rapamycin resistance and that, in turn, TORC1

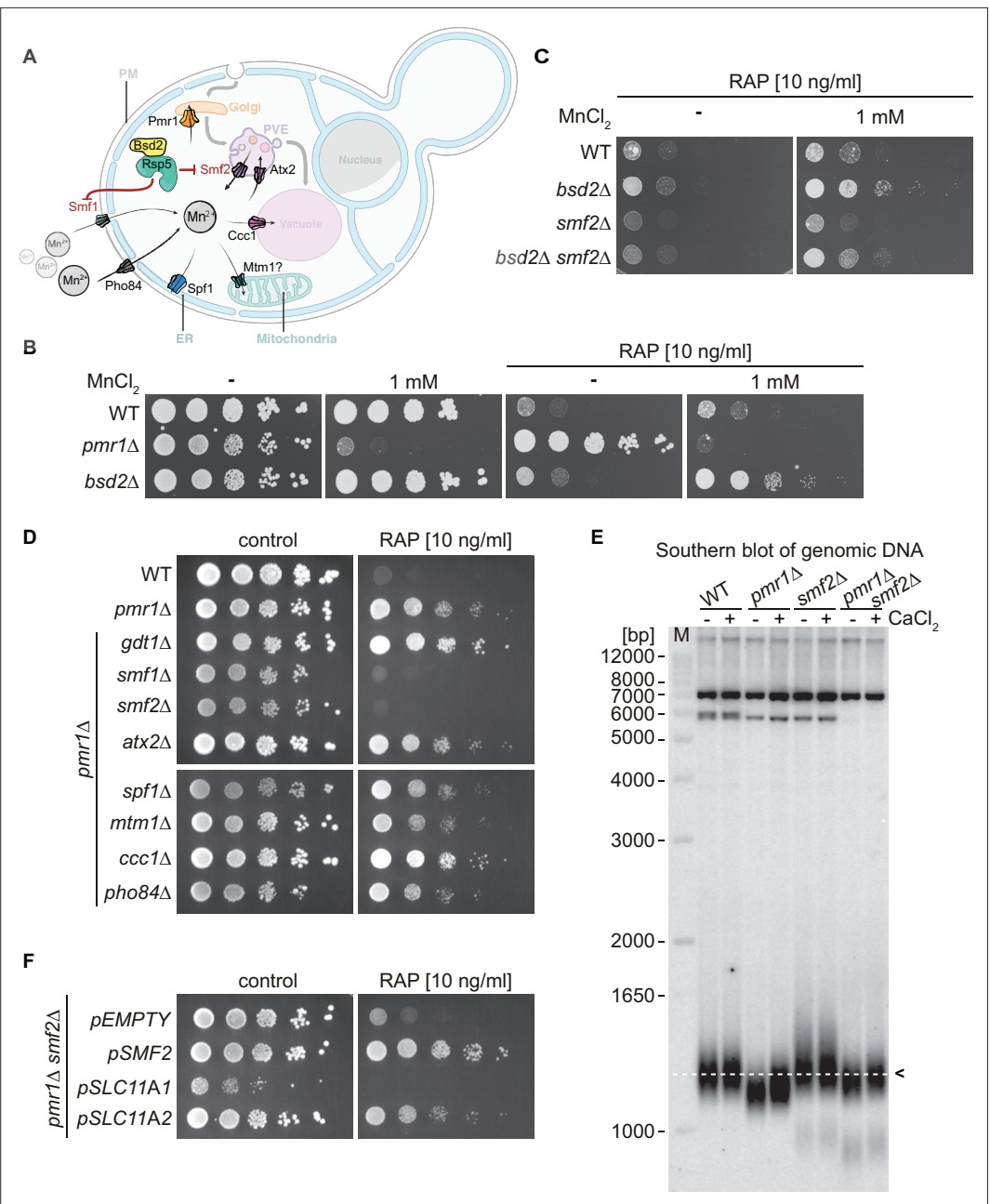

**Figure 1.** Natural resistance-associated macrophage protein (NRAMP) transporters link Mn$^{2+}$-import to rapamycin resistance. (**A**) Schematical outline of yeast Mn$^{2+}$ transporters and their intracellular localization. PM, plasma membrane; PVE, pre-vacuolar endosomes; ER, endoplasmic reticulum. Note that Bsd2 is a specific adaptor protein for Rsp5-mediated Smf1 and Smf2 ubiquitination in response to Mn$^{2+}$ overload. The Golgi Mn$^{2+}$ transporter Gdt1 is omitted for clarity. (**B–D**) Growth on MnCl$_2$ and/or rapamycin-containing medium (RAP). Ten-fold dilutions of exponentially growing cells are shown. Strains and compound concentrations are indicated. Note that the medium used in (**D**) was supplemented with 10 mM CaCl$_2$. Data obtained in a medium without CaCl$_2$ are shown in *Figure 1—figure supplement 3*. (**E**) Southern blot analysis of telomere length. Genomic DNA was derived from cells grown in a medium supplemented or not with CaCl$_2$ and cleaved by *Xho*I before agarose gel electrophoresis. The 1.3 kb average length of telomeres from WT cells (dashed white line, black arrow) and size marker (**M**) are shown. (**F**) Growth of *pmr1Δ smf2Δ* double mutants transformed with plasmids expressing yeast Smf2, *Mus musculus* SLC11A1, or SLC11A2 on rapamycin-containing medium. Complementary data showing that *pmr1Δ* rapamycin resistance is not linked to Gap1 or Tor1 localization is provided in *Figure 1—figure supplement 1*. Rapamycin sensitivity of *bsd2Δ* cells overexpressing the Vcx1-M1 transporter are shown in *Figure 1—figure supplement 2*.

*Figure 1 continued on next page*

*Figure 1 continued*

The online version of this article includes the following source data and figure supplement(s) for figure 1:

**Source data 1.** Uncropped autoradiography image shown in *Figure 1E*.

**Source data 2.** Raw autoradiography image shown in *Figure 1E*.

**Figure supplement 1.** *pmr1Δ* rapamycin resistance is not linked to Gap1 or Tor1 localization.

**Figure supplement 1—source data 1.** Uncropped blot shown in *Figure 1—figure supplement 1C*.

**Figure supplement 1—source data 2.** Raw blot shown in *Figure 1—figure supplement 1C*.

**Figure supplement 2.** Growth of *bsd2Δ* mutants expressing Vcx1-M1 from plasmid *pVCX1-M1* on $MnCl_2$ and/or rapamycin (RAP) containing medium.

**Figure supplement 3.** Natural resistance-associated macrophage protein (NRAMP) transporters mediate rapamycin resistance of *pmr1Δ* mutants.

inhibition by rapamycin regulates NRAMP transporter availability. Moreover, intracellular $Mn^{2+}$ excess antagonizes rapamycin-induced autophagy, mitophagy, and Rtg1-3 transcription factor complex-dependent retrograde response activation. Surprisingly, our in vitro analyses reveal that TORC1 protein kinase activity is strongly activated in the presence of $MnCl_2$. In our attempts to understand the mechanisms underlying these observations, we discovered that $Mn^{2+}$, when compared to $Mg^{2+}$, significantly boosts the affinity of TORC1 for ATP. Combined, our findings also indicate that TORC1 activity is regulated by and regulates intracellular $Mn^{2+}$ levels, defining $Mn^{2+}$ homeostasis as a key factor in cell growth control. Importantly, our studies in human cells indicate that $Mn^{2+}$-driven TORC1 activation is likely conserved throughout evolution, opening new perspectives for our understanding of $Mn^{2+}$ toxicities and their role in neurodegenerative disorders and aging.

## Results

### NRAMP transporters regulate cytoplasmic $Mn^{2+}$ levels and rapamycin resistance

Yeast cells lacking the Golgi-localized P-type ATPase Pmr1, which transports $Ca^{2+}$ and $Mn^{2+}$ ions from the cytoplasm to the Golgi lumen, are resistant to the TORC1 inhibitor rapamycin (*Devasahayam et al., 2006*; *Devasahayam et al., 2007*), a phenotype that is generally associated with increased TORC1 activity. In *pmr1Δ* mutants, defective $Mn^{2+}$ shuttling at the Golgi leads to protein sorting defects and accumulation of the general amino acid permease Gap1 at the plasma membrane (*Kaufman et al., 1994*). In theory, this may translate into unrestrained uptake and intracellular accumulation of amino acids, and thus hyperactivation of TORC1. However, arguing against such a model, we found cells lacking both Pmr1 and Gap1 to remain resistant to low doses of rapamycin (*Figure 1—figure supplement 1A*). We then asked whether loss of Pmr1 may affect the expression levels or cellular localization of TORC1. Our results indicated that GFP-Tor1 protein levels and localization to vacuolar and endosomal membranes remained unaltered in exponentially growing and rapamycin-treated WT and *pmr1Δ* cells (*Figure 1—figure supplement 1B and C*). Given the roughly fivefold increased intracellular $Mn^{2+}$ levels in cells lacking Pmr1 (*Lapinskas et al., 1995*), we then considered the possibility that $Mn^{2+}$ may have a more direct role in TORC1 activation. We, therefore, assessed the rapamycin sensitivity of cells lacking the adaptor protein Bsd2, which mediates Rsp5-dependent degradation in response to high $Mn^{2+}$ levels of both the plasma membrane- and endosomal membrane-resident NRAMP $Mn^{2+}$ transporters Smf1 and Smf2, respectively (*Figure 1A*; *Liu et al., 1997*). Interestingly, *bsd2Δ* cells were as sensitive to rapamycin as WT cells, but, unlike WT cells, could be rendered rapamycin resistant by the addition of 1 mM $MnCl_2$ in the growth medium (*Figure 1B* and *Figure 1—figure supplement 2*). This effect was strongly reduced in the absence of Smf2 (*Figure 1C*), suggesting that Smf2-dependent endosomal $Mn^{2+}$ export and, consequently, cytoplasmic accumulation of $Mn^{2+}$ may be required for rapamycin resistance under these conditions. In line with such a model, we found that overexpression of the vacuolar membrane-resident Vcx1-M1 transporter, which imports $Mn^{2+}$ into the vacuolar lumen, suppressed the $Mn^{2+}$-induced rapamycin resistance of *bsd2Δ* mutants and increased their sensitivity to rapamycin in the absence of extracellular $MnCl_2$ supply (*Figure 1—figure supplement 2*).

As schematically outlined in *Figure 1A*, several metal transporters have been associated with $Mn^{2+}$ transport in yeast, including those localized at the plasma membrane (Smf1 and Pho84), the endosomes (Smf2 and Atx2), the Golgi (Pmr1 and Gdt1), the vacuole (Ccc1 and Vcx1), the ER (Spf1), and possibly the mitochondria (Mtm1). To identify which of these transporters contributes to the rapamycin resistance of *pmr1Δ* cells, we next monitored the growth of double mutants lacking Pmr1 and any of the corresponding metal transporters in the presence of rapamycin (*Figure 1D*). Interestingly, only loss of Smf1 or Smf2 reestablished rapamycin sensitivity in *pmr1Δ* cells, while loss of Gdt1 or Ccc1 even slightly enhanced the rapamycin resistance phenotype of *pmr1Δ* cells. Of note, growth was monitored on $CaCl_2$-supplemented media that improved the growth of *pmr1Δ* cells lacking Smf1 or Smf2 without affecting their sensitivity to rapamycin, which also rules out the possibility that $Ca^{2+}$ is associated with the observed rapamycin resistance phenotype (*Figure 1D* and *Figure 1—figure supplement 3*). Taken together, our data indicate that increased intracellular $Mn^{2+}$ levels in *pmr1Δ* cells lead to rapamycin resistance and that this phenotype can be suppressed either by reducing $Mn^{2+}$ import through loss of Smf1 or, possibly, by increasing $Mn^{2+}$ sequestration in endosomes through loss of Smf2 as previously suggested (*Luk and Culotta, 2001*). To confirm our prediction that loss of Smf2 reduces cytoplasmic $Mn^{2+}$ levels, we measured $MnCl_2$-dependent telomere length shortening as an indirect proxy for cytoplasmic and nuclear Mn levels (*García-Rodríguez et al., 2015*). Accordingly, telomere length was significantly decreased in *pmr1Δ* cells (when compared to WT cells), increased in *smf2Δ cells*, and similar between *pmr1Δ smf2Δ* and WT cells (*Figure 1E*). These data, therefore, corroborate our assumption that loss of Smf2 suppresses the high cytoplasmic and nuclear $Mn^{2+}$ levels of *pmr1Δ* cells.

The function of metal transporters is highly conserved across evolution as exemplified by the fact that the expression of the human Pmr1 ortholog, the secretory pathway $Ca^{2+}/Mn^{2+}$ ATPase ATP2C1/SPCA1 (ATPase secretory pathway $Ca^{2+}$ transporting 1), can substitute for Pmr1 function in yeast (*Muncanovic et al., 2019*). A similar degree of functional conservation from lower to higher eukaryotes exists for NRAMP transporters (*Sacher et al., 2000*). Because Smf2 is of specific interest in the context of the present study, we asked whether the orthologous mouse proteins, that is, the divalent metal transporter SLC11A1 (NRAMP1) and SLC11A2 (DMT1) isoforms (solute carrier family 11 member 1 and 2, respectively), can restore rapamycin resistance in *pmr1Δ smf2Δ* double mutants. Expression of SLC11A1 did not rescue the lack of Smf2, leading to poor growth even in the absence of rapamycin. However, the SLC11A2 isoform complemented Smf2 function in these assays (*Figure 1F*), indicating that Pmr1 and Smf2 are evolutionarily conserved transporters that are required for $Mn^{2+}$ homeostasis.

## Elevated levels of intracellular $Mn^{2+}$ antagonize rapamycin-induced autophagy, mitophagy, and Rtg1-3 retrograde signaling

Growth inhibition by rapamycin mimics starvation conditions and leads to the degradation and recycling of a wide spectrum of biological macromolecules via autophagy. In this context, the *pmr1Δ* mutant has previously been found to be defective in nutrient depletion-induced mitophagy (*Kanki et al., 2010*). We thus wondered if rapamycin-induced autophagic processing may also be defective in *pmr1Δ* mutants. We took advantage of a GFP-Atg8 fusion construct (*Cheong and Klionsky, 2008*) to monitor autophagy through GFP-Atg8 synthesis and processing in cells that had been subjected to rapamycin treatment for up to 6 hr. Rapamycin-induced GFP-Atg8 expression was strongly reduced in *pmr1Δ* cells and this phenotype was mitigated in the *pmr1Δ smf2Δ* double mutant (*Figure 2A and B*), suggesting that the initiation of autophagy is compromised by elevated cytoplasmic $Mn^{2+}$ levels. Next, we assessed mitophagy by following the expression and degradation of the GFP-tagged mitochondrial membrane protein Om45 (*Kanki et al., 2009*). Rapamycin treatment led to a time-dependent up-regulation of Om45-GFP protein levels in all tested strains (*Figure 2C and D*). However, the accumulation of the cleaved GFP protein was strongly reduced in the *pmr1Δ* single mutant, while this effect was again suppressed in the *pmr1Δ smf2Δ* double mutant. Combined, our results therefore suggest that the intracellular $Mn^{2+}$ flux modulates both rapamycin-induced autophagy and mitophagy.

To identify additional responses to elevated cytosolic $Mn^{2+}$ levels, we took advantage of our previously published transcriptome analysis of *pmr1Δ* cells (*García-Rodríguez et al., 2015*). Careful analysis of these datasets revealed that the expression levels of genes activated by the heterodimeric Rtg1-Rtg3 transcription factor are reduced in *pmr1Δ* cells (see *Supplementary file 1*). Since TORC1

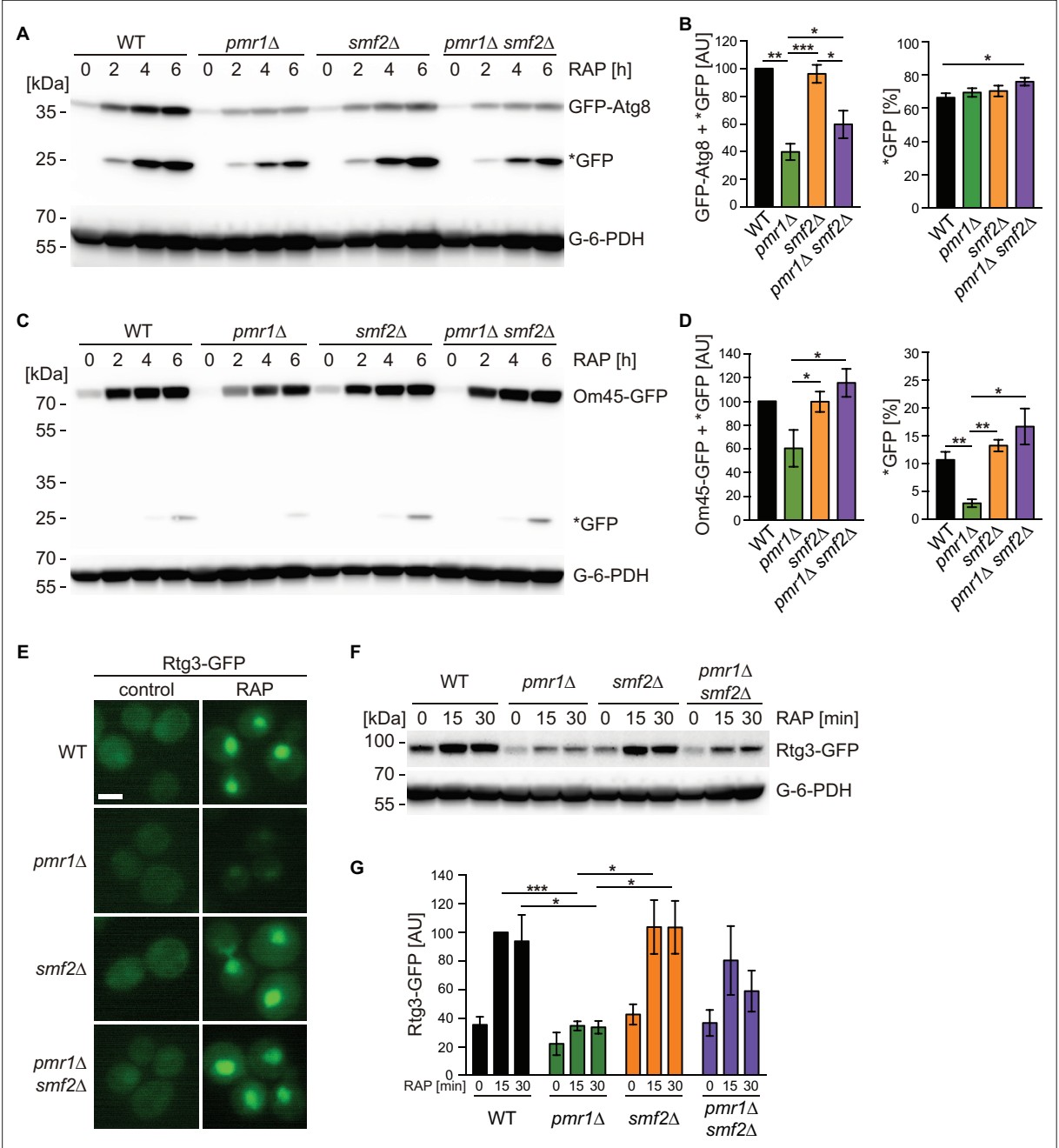

**Figure 2.** Intracellular manganese (Mn) excess antagonizes rapamycin-induced autophagy, mitophagy, and Rtg1-3 retrograde signaling.
(**A**) Exponentially growing WT and indicated mutant strains expressing plasmid-encoded GFP-Atg8 were treated for up to 6 hr with 200 ng/ml rapamycin (RAP). GFP-Atg8 and cleaved GFP (*GFP) protein levels were analyzed by immunoblotting. Glucose-6-phosphate dehydrogenase (G-6-PDH) levels were used as a loading control. (**B**) Quantification of GFP-Atg8 and *GFP levels after a 6 hr rapamycin treatment. Total GFP signal (Atg8-GFP + *GFP) normalized to WT levels (left) and percentage of *GFP relative to the total GFP signal (right) are plotted. Data represent means ± SEM of independent experiments (n=4). Statistical analysis: two-tailed t-test (paired for normalized data, unpaired for GFP* percentage). *p<0.05; **p<0.01; ***p<0.001. (**C**) Exponentially growing WT and indicated mutant strains expressing Om45-GFP from the endogenous locus were treated and processed as in (**A**). (**D**) Quantification of Om45-GFP and *GFP levels after a 6 hr rapamycin treatment. Details as in (**B**) with n=3. (**E**) Representative fluorescence microscope images of WT, *pmr1Δ*, *smf2Δ*, and *pmr1Δ smf2Δ* mutants expressing an episomic Rtg3-GFP reporter construct. Exponentially growing cells were treated or not (control) for 30 min with 200 ng/ml rapamycin (RAP). Scale bar represents 5 μm. (**F**) Rtg3-GFP expressing cells were treated for up to 30 min with 200 ng/ml rapamycin. Protein levels were analyzed by immunoblotting. G-6-PDH levels were used as a loading control. (**G**) Quantification of Rtg3-GFP signals. Values normalized to WT levels after 15 min of rapamycin treatment (highest signal) are plotted. Data represent means ± SEM of independent experiments (n=4). Statistical analysis: paired two-tailed t-test.

*Figure 2 continued on next page*

*Figure 2 continued*

The online version of this article includes the following source data for figure 2:

**Source data 1.** Quantification of blots for graphs shown in *Figure 2B–D–G.*

**Source data 2.** Uncropped blots shown in *Figure 2A–C–F* and quantified in *Figure 2B–D.*

**Source data 3.** Raw blots shown in *Figure 2A–C–F* and quantified in *Figure 2B–D.*

inhibits cytoplasmic-to-nuclear translocation of Rtg1-Rtg3 (*Ruiz-Roig et al., 2012*), we monitored the localization and protein levels of Rtg3-GFP in exponentially growing and rapamycin-treated WT and *pmr1Δ* cells. Rapamycin treatment not only induced nuclear enrichment of Rtg3-GFP as reported (*Figure 2E*; *Ruiz-Roig et al., 2012*) but also significantly increased the levels of Rtg3-GFP (*Figure 2F and G*). Loss of Pmr1, in contrast, significantly reduced the Rtg3-GFP levels in exponentially growing and rapamycin-treated cells, which also translated into barely visible levels of Rtg3-GFP in the nucleus (*Figure 2E–G*). Importantly, and in line with our finding that loss of Smf2 suppresses the high cytoplasmic and nuclear $Mn^{2+}$ levels of *pmr1Δ* cells (see above), these latter defects in *pmr1Δ* cells were partially suppressed by loss of Smf2. Thus, our findings posit a model in which elevated cytoplasmic $Mn^{2+}$ levels antagonize autophagy, mitophagy, and Rtg1-3-dependent retrograde signaling presumably through activation of TORC1.

## $MnCl_2$ stimulates TORC1 kinase activity in vitro

The yeast Tor1 kinase is a member of the phosphatidylinositol 3-kinase (PI3K)-related kinase (PIKK) family that can phosphorylate the human eukaryotic translation initiation factor 4E binding protein (eIF4-BP/PHAS-I) in vitro (*Alarcon et al., 1999*). Curiously, it does so much more efficiently when the respective in vitro kinase assays contain $Mn^{2+}$ rather than $Mg^{2+}$ as the sole divalent cation, a property that it appears to share with PI3-kinases (*Carpenter et al., 1993*; *Dhand et al., 1994*; *Foukas et al., 2004*). A similar preference for $Mn^{2+}$ over $Mg^{2+}$ has also been observed in mTOR kinase autophosphorylation assays (*Brunn et al., 1996*; *Withers et al., 1997*). Based on these and our observations, we decided to assess whether $Mn^{2+}$ may act as a metal cofactor for TORC1 activity in vitro using TORC1 purified from yeast and a truncated form of Lst4 (Lst4$^{Loop}$; *Nicastro et al., 2021*) as a substrate. In control experiments without divalent ions, TORC1 activity remained undetectable (*Figure 3A and B*). The addition of $MnCl_2$, however, not only stimulated TORC1 in vitro in a concentration-dependent manner, but also activated TORC1 dramatically more efficiently than $MgCl_2$ (with 25-fold lower levels of $MnCl_2$ [38 µM] than $MgCl_2$ [980 µM] promoting half-maximal activation of TORC1; *Figure 3A and B*). We next considered the possibility that $Mn^{2+}$ is superior to $Mg^{2+}$ in favoring the coordination of ATP in the catalytic cleft or TORC1. Supporting this idea, we found that $Mn^{2+}$ significantly reduced the $K_m$ for ATP (5.3-fold) of TORC1 (*Figure 3C and D*). Moreover, even in the presence of saturating $Mg^{2+}$ levels (i.e. 4 mM), the addition of 160 µM $Mn^{2+}$ was able to enhance the $V_{MAX}$ almost twofold and decrease the $K_m$ for ATP of TORC1 from 50.7 to 34.4 µM, which indicates that $Mn^{2+}$ can efficiently compete with $Mg^{2+}$ and thereby activate TORC1. Notably, the conditions used in our in vitro kinase assays are quite comparable to the in vivo situation: accordingly, intracellular $Mg^{2+}$ levels in yeast are approximately around 2 mM (*van Eunen et al., 2010*), while the $Mn^{2+}$ levels range from 26 µM in WT cells to 170 µM in *pmr1Δ* cells (*McNaughton et al., 2010*). Our in vitro assays, therefore, provide a simple rationale for why *pmr1Δ* cells are resistant to rapamycin: elevated $Mn^{2+}$ levels in *pmr1Δ* favorably boost the kinetic parameters of TORC1.

## TORC1 regulates NRAMP transporter protein levels

To maintain appropriate intracellular $Mn^{2+}$ concentrations, cells adjust Smf1 and Smf2 protein levels through Bsd2-dependent and -independent, post-translational modifications that ultimately trigger their vacuolar degradation through the endocytic multivesicular body pathway (*Liu and Culotta, 1994*; *Liu et al., 1997*). Because NRAMP transporters are important for Mn-dependent TORC1 activation, and because TORC1 is often embedded in regulatory feedback loops to ensure cellular homeostasis (*Eltschinger and Loewith, 2016*), we next asked whether rapamycin-mediated TORC1 inactivation may affect the levels and/or localization of Smf1 and Smf2 using a strain that expresses N-terminally tagged Smf1 (GFP-Smf1) under its own promoter from a non-endogenous locus or a strain that expresses C-terminally tagged Smf2 (Smf2-GFP) from its endogenous locus (*García-Rodríguez*

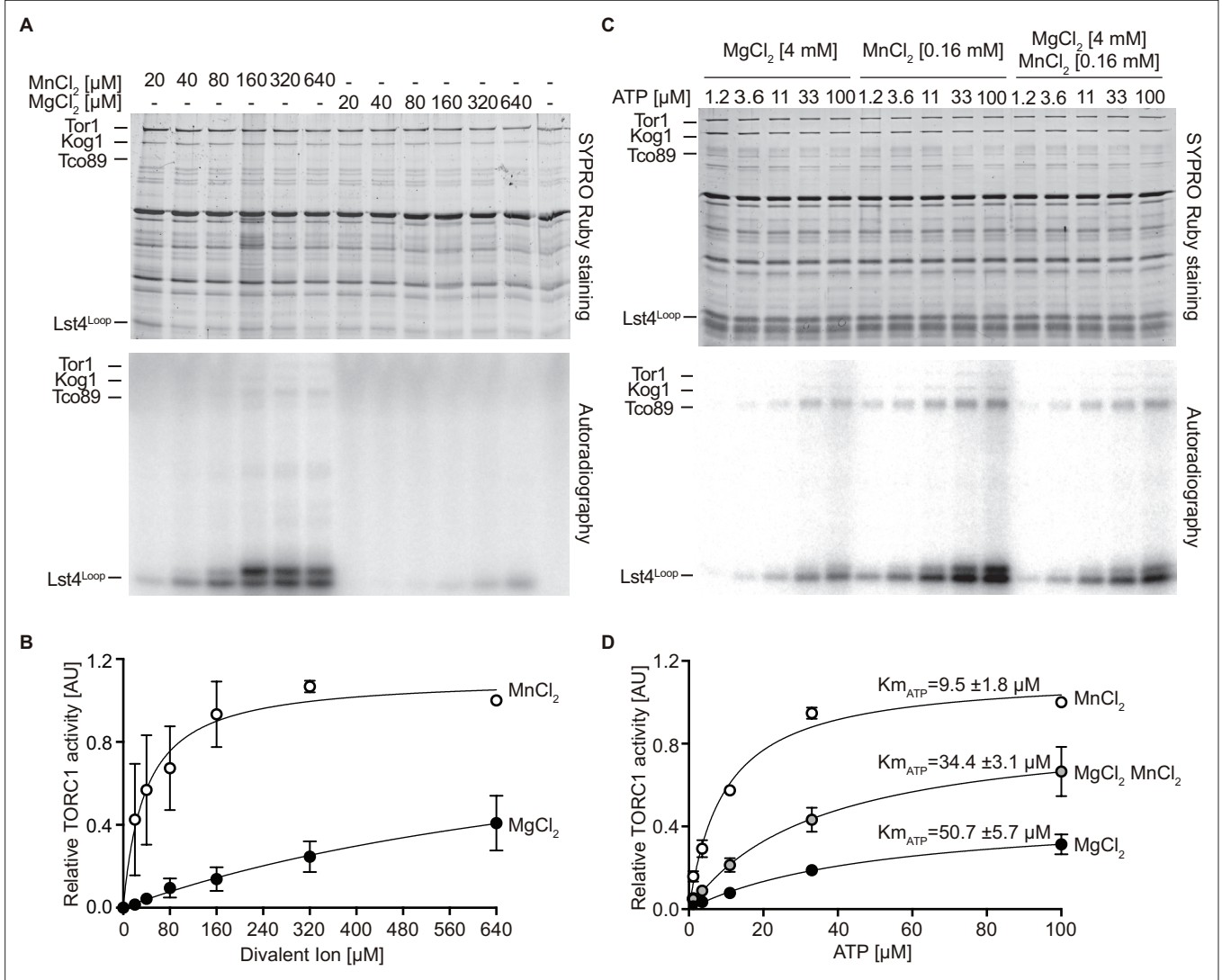

**Figure 3.** MnCl$_2$ stimulates TORC1 kinase activity in vitro and in vivo. (**A**) In vitro TORC1 kinase assays using [γ-$^{32}$P]-ATP, recombinant Lst4$^{Loop}$ as substrate, and increasing concentrations (twofold dilutions) of MgCl$_2$ or MnCl$_2$. Substrate phosphorylation was detected by autoradiography (lower blot) and SYPRO Ruby staining is shown as loading control (upper blot). (**B**) Quantification of the assay shown in (**A**). Curve fitting and parameter calculations were performed with GraphPad Prism. Data shown are means (± SEM, n=3). (**C**) In vitro kinase assays (as in A) using the indicated concentrations of MgCl$_2$ and/or MnCl$_2$ and increasing concentrations of ATP. Substrate phosphorylation was detected by autoradiography (lower blot) and SYPRO Ruby staining is shown as loading control (upper blot). (**D**) Quantification of the assay shown in (**C**). Data shown are means (± SEM, n=3). Curve fitting and parameter calculations were performed with GraphPad Prism. Km$_{ATP}$ are shown for each curve. V$_{MAX}$ [MnCl$_2$]=1.13 ± 0.06, V$_{MAX}$ [MgCl$_2$ MnCl$_2$]=0.89 ± 0.03, V$_{MAX}$ [MgCl$_2$]=0.47 ± 0.02.

The online version of this article includes the following source data for figure 3:

**Source data 1.** Quantification of autoradiographies for graphs shown in *Figure 3B–D*.

**Source data 2.** Uncropped gels and autoradiographies shown in *Figure 3A–C* and quantified in *Figure 3B–D*.

**Source data 3.** Raw gels and autoradiographies shown in *Figure 3A–C* and quantified in *Figure 3B–D*.

---

*et al., 2015*; *Renz et al., 2020*). Rapamycin treatment triggered a strong increase in GFP-Smf1 levels (*Figure 4A and B*), which is likely due to transcriptional activation of Smf1 under these conditions as published earlier (*Reinke et al., 2006*). In line with this interpretation, we found the respective increase to be abolished when rapamycin-treated cells were co-treated with the protein synthesis inhibitor cycloheximide (CHX). In addition, because GFP-Smf1 appeared to be degraded at a similar rate in CHX-treated cells that were treated, or not, with rapamycin, our data further indicate that TORC1 does not control Smf1 levels through post-translational control of Smf1 turnover. Interestingly,

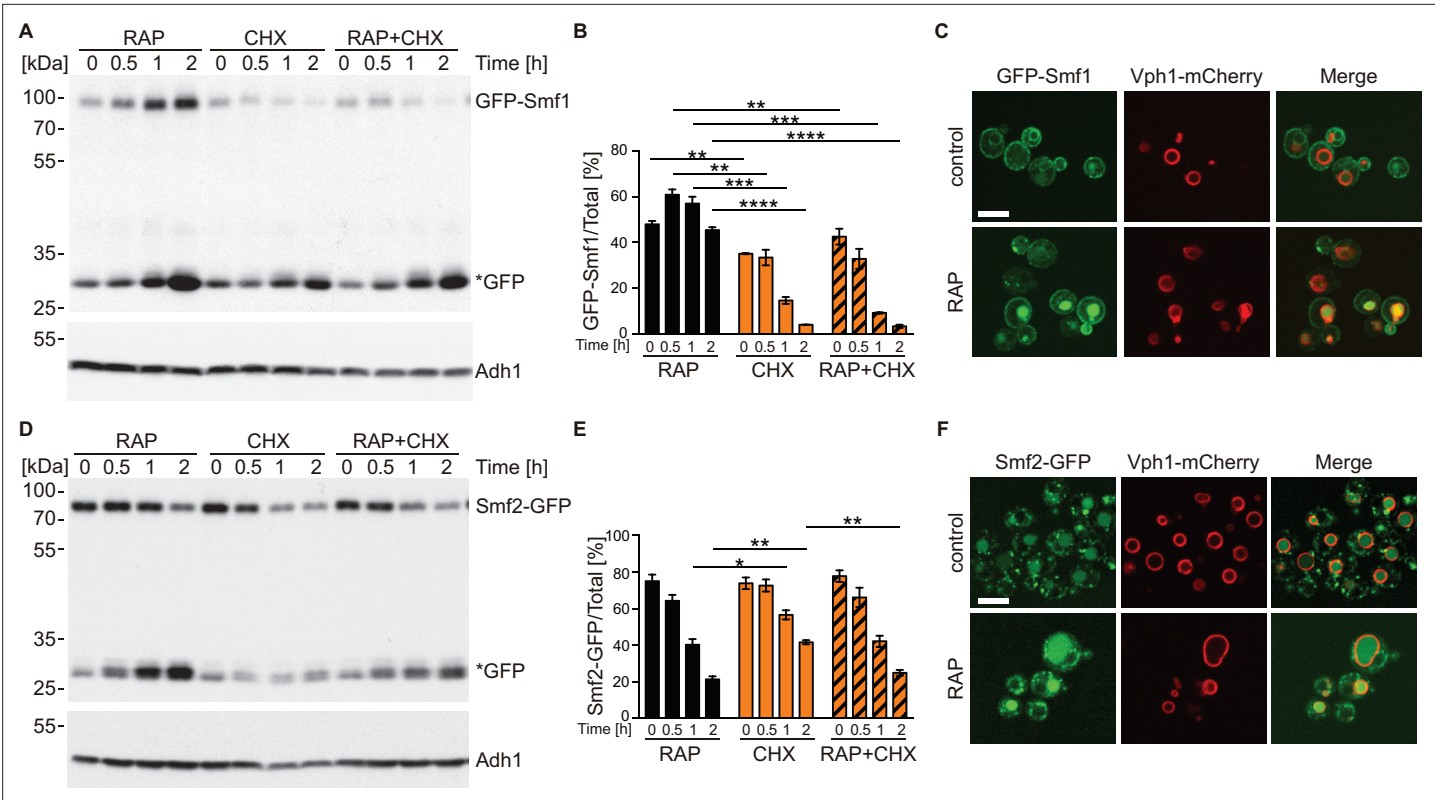

**Figure 4.** TORC1 regulates natural resistance-associated macrophage protein (NRAMP) transporter levels. (**A**) GFP-Smf1 expressing cells were cultivated for 5–6 hr in a synthetic medium devoid of manganese sulfate before being treated with 200 ng/ml rapamycin (RAP), 25 µg/ml cycloheximide (CHX), or both compounds (RAP + CHX) for the indicated times. GFP-Smf1 and cleaved GFP (*GFP) protein levels were analyzed by immunoblotting using an anti-GFP antibody. Alcohol dehydrogenase (Adh1) protein levels, probed with anti-Adh1 antibodies, served as a loading control. (**B**) Quantification of GFP-Smf1. Percentages of GFP-Smf1 relative to the total GFP signal (GFP-Smf1 + GFP) are plotted. Data represent means ± SEM of independent experiments (n=3). Statistical analysis: unpaired two-tailed t-test. *p<0.05; **p<0.01, ***p<0.001, ****p<0.0001. (**C**) Microscopic analysis of GFP-Smf1 localization. Cells co-expressing GFP-Smf1 and the vacuolar marker Vph1-mCherry were grown exponentially in a manganese-free medium for 5–6 hr, then treated with 200 ng/ml rapamycin for 2 hr. Scale bar represents 5 µm. (**D–E**) Cells expressing Smf2-GFP from its endogenous locus were grown, treated, and processed as in (**A**). (**F**) Microscopic analysis of Smf2-GFP localization. Cells co-expressing Smf2-GFP and the vacuolar marker Vph1-mCherry were cultivated, treated, and examined as in (**C**).

The online version of this article includes the following source data for figure 4:

**Source data 1.** Quantification of blots for graphs shown in *Figure 4B–E*.

**Source data 2.** Uncropped blots shown in *Figure 4A–D* and quantified in *Figure 4B–E*.

**Source data 3.** Raw blots shown in *Figure 4A–D* and quantified in *Figure 4B–E*.

GFP-Smf1 remained predominantly at the plasma membrane in rapamycin-treated cells, even though these cells also accumulated more cleaved GFP in the vacuoles (*Figure 4C*). We infer from these data that TORC1 inhibition activates Smf1 expression, likely as part of a feedback control loop through which TORC1 couples its activity to Mn$^{2+}$ uptake. Rapamycin-treated cells that were either co-treated or not with CHX exhibited Smf2-GFP levels that steadily decreased at a similar rate, with cleaved GFP accumulating in parallel (*Figure 4D and E*). Since this rate appeared to be higher than the one observed in cells treated with CHX alone, our data suggest that TORC1 antagonizes the turnover of Smf2. This was further corroborated by our fluorescence microscopy analyses revealing that the GFP signal in Smf2-GFP-expressing cells shifted from late Golgi/endosomal foci (see *García-Rodríguez et al., 2015*) to a predominant signal within the vacuolar lumen when cells were treated with rapamycin (*Figure 4F*). Whether this event is an indirect consequence of higher Mn$^{2+}$ uptake under these conditions (see above), or potentially part of a local feedback control mechanism of endosomal TORC1 (*Hatakeyama et al., 2019*), remains to be addressed in future studies.

## Mn²⁺ activates mTORC1 signaling in human cells

To assess whether the response of TORC1 to $Mn^{2+}$ is conserved among eukaryotes, we investigated if externally supplied $MnCl_2$ activates mTORC1 signaling in mammalian cells. For this purpose, we used the human cell lines U2OS and HEK293T, which are widely used for the cellular dissection of mTORC1-mediated mechanisms in response to nutritional inputs (**Bodineau et al., 2021**). First, we examined the sufficiency of $Mn^{2+}$ to maintain the activity of the mTORC1 pathway by incubating U2OS cells in an amino acid starvation medium supplemented with increasing amounts of $MnCl_2$ (0–1 mM) for 2 hr. mTORC1 activity was assessed by monitoring phosphorylation of the mTORC1 downstream targets RPS6KB (ribosomal protein S6 kinase B; phospho-Threonine³⁸⁹) and RPS6 (ribosomal protein S6; phospho-Serine²³⁵⁻²³⁶). As shown in *Figure 5A*, and as expected, both RPS6KB and RPS6 were fully phosphorylated at those specific residues in cells incubated in the presence of amino acids and completely de-phosphorylated in cells incubated in the absence of amino acids, without $MnCl_2$. In agreement with a positive action of $Mn^{2+}$ toward mTORC1 in human cells, we observed a robust and dose-dependent increase in both RPS6KB and RPS6 phosphorylation in amino acid-starved cells incubated in the presence of $MnCl_2$ at concentrations higher than 0.05 mM, with maximum phosphorylation observed at 0.5 mM of $MnCl_2$. Confirming this result, we observed a similar response of RPS6KB and RPS6 phosphorylation to $MnCl_2$ treatment in HEK293T cells, except that the maximum response was reached at 1 mM (*Figure 5B*). These results indicate that $Mn^{2+}$ is sufficient to maintain the activity of the mTORC1 pathway in the absence of amino acid inputs, at least during short periods (2 hr). To corroborate this conclusion, we also tested the phosphorylation of additional downstream targets of mTORC1 in response to a fixed concentration of $MnCl_2$ (0.35 mM) in U2OS cells. Amino acid starvation of U2OS cells with $MnCl_2$ was sufficient to maintain the phosphorylation of the downstream targets of mTORC1 RPS6KB, RPS6, EIF4EBP1, and ULK1 (unc-51 like autophagy activating kinase 1) (*Figure 5C*). In line with our finding that $MnCl_2$ retained AKT1 (AKT serine/threonine kinase 1) phosphorylation in amino acid-starved cells, $Mn^{2+}$ has been reported to activate the mTORC1 upstream kinase AKT1, which is also a known mTORC2 downstream target (**Bryan et al., 2018**; **Gaubitz et al., 2016**). However, AKT1 phosphorylation appeared to be much less pronounced than the phosphorylation of the mTORC1 downstream targets, validating the specificity of Mn toward mTORC1. Whether $Mn^{2+}$ may stimulate TORC2 activity remains to be explored.

Finally, to assess the physiological relevance of mTORC1 activation in response to $Mn^{2+}$, we analyzed the effect of $MnCl_2$ treatment in autophagy, a cellular process inhibited by TORC1 both in yeast and in mammalian cells (**Noda and Ohsumi, 1998**; **Villar et al., 2017**). To this end, we analyzed the autophagic marker MAP1LC3I/II (microtubule-associated protein 1 light chain 3) (**Klionsky et al., 2016**). During autophagy initiation, MAP1LC3I is lipidated, thereby increasing the levels of MAP1LC3II. As previously reported, the withdrawal of amino acids led to a rapid increase in MAP1LC3II levels, indicating an increase in autophagy initiation (*Figure 5D*). Of note, the addition of $MnCl_2$ completely abolished the increase in MAP1LC3II levels, thus confirming that $Mn^{2+}$ prevented the initiation of autophagy downstream of mTORC1. These results were also observed in cells incubated in the presence of chloroquine, an inhibitor of the fusion of autophagosomes with the lysosome, thus confirming that $Mn^{2+}$ influences autophagy initiation, the process controlled by mTORC1 (*Figure 5E*). In addition, we analyzed the aggregation of MAP1LC3 upon autophagosome formation in U2OS cells stably expressing a GFP-MAP1LC3 construct. GFP aggregation indicates autophagosome formation in these cells. Similar to what we observed with endogenous MAP1LC3, GFP aggregation was increased during amino acid withdrawal and this increase was completely abolished by $MnCl_2$ treatment (*Figure 5F–G*). This result further confirms that the Mn-dependent activation of mTORC1 in human cells is physiologically relevant for autophagy inhibition. Altogether, our results show that $Mn^{2+}$ is sufficient to keep mTORC1 activated in the absence of other inputs in human cells and to inhibit autophagy downstream of mTORC1. These findings indicate that $Mn^{2+}$-mediated activation of TORC1 is evolutionarily conserved from yeast to humans.

## Discussion

Studies with yeast lacking the divalent Ca/Mn ion Golgi transporter Pmr1, which displays an increase in intracellular $Mn^{2+}$ levels due to impaired $Mn^{2+}$ detoxification (**McNaughton et al., 2010**), have pinpointed a functional link between $Mn^{2+}$ homeostasis and TORC1 signaling (**Devasahayam et al.,**

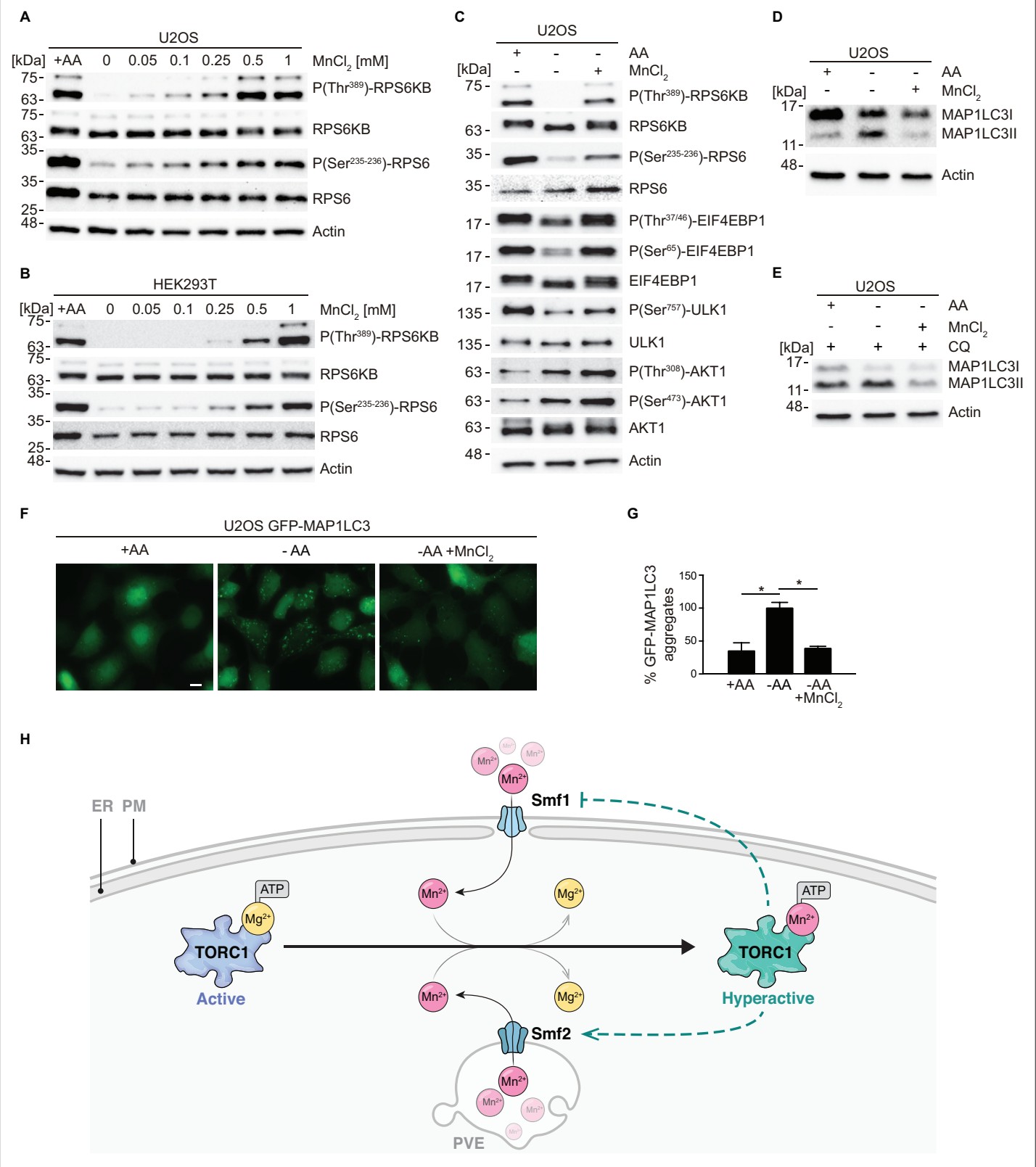

**Figure 5.** MnCl₂ maintains mTORC1 activity during starvation in human cells. (**A–B**) Human U2OS (**A**) and HEK293T (**B**) cells were starved for all amino acids and supplemented with increasing concentrations of MnCl₂ as indicated for 2 hr. Phosphorylation of the mTORC1 downstream targets RPS6KB (ribosomal protein S6 kinase B) and RPS6 (ribosomal protein S6) was assessed by immunoblot analysis. A control with cells incubated in the presence of all proteinogenic amino acids (+AA) was included as a positive control. (**C**) Human U2OS were starved for all amino acids (-AA) and supplemented

*Figure 5 continued on next page*

*Figure 5 continued*

with 0.35 mM of MnCl$_2$ for 2 hr. Phosphorylation of RPS6KB, RPS6, EIF4EBP1, ULK1, and AKT1 was assessed by immunoblot analysis. A control with cells incubated in the presence of all proteinogenic amino acids (+AA) was included as a positive control. (**D–E**) Human U2OS cells were treated as in (**C**) for 4 hr either in the absence (**D**) or the presence (**E**) of chloroquine (CQ). Autophagic marker MAP1LC3I/II was then analyzed by immunoblot. (**F–G**) GFP-MAP1LC3 expressing U2OS cells were incubated as in (**D**). GFP-MAP1LC3 aggregation was assessed by confocal microscopy (**F**) and quantified using ImageJ software (**G**). Scale bar indicates 10 µm. Values were normalized to -AA condition. Graphs represent mean ± SD, with n=3 (*p<0.05, ANOVA followed by Tukey's test). (**H**) Model of Mn$^{2+}$-driven TORC1 activation. The Smf1/2 natural resistance-associated macrophage protein (NRAMP) transporter-dependent increase in cytoplasmatic Mn$^{2+}$ levels favors TORC1-Mn$^{2+}$ binding and ATP coordination, leading to TORC1 hyperactivation. NRAMP transporters are part of a feedback control mechanism impinged by TPRC1 (dashed lines).

The online version of this article includes the following source data for figure 5:

**Source data 1.** Quantification of microscopy images for graph shown in *Figure 5G*.

**Source data 2.** Uncropped blots shown in *Figure 5A–E*.

**Source data 3.** Raw blots shown in *Figure 5A–E*.

*2006*; *Devasahayam et al., 2007*). However, the underlying mechanism(s) of how Mn$^{2+}$ impinges on TORC1 has so far remained elusive. Our study identifies Mn$^{2+}$ as a divalent metal cofactor that stimulates the enzymatic activity of the TORC1 complex in vitro and suggests that Mn$^{2+}$ functions similarly in vivo both in yeast and in mammalian cells. Our results support a model in which the PI3K-related protein kinase Tor1 is activated by Mg$^{2+}$, but requires Mn$^{2+}$ as a metal cofactor for maximal activity as this allows it to better coordinate ATP (*Figure 5H*). While it seemingly shares this feature with PI3Ks (*Carpenter et al., 1993*; *Dhand et al., 1994*; *Foukas et al., 2004*), serine/threonine protein kinases are generally known to be preferably activated by Mg$^{2+}$ rather than Mn$^{2+}$ (*Knape et al., 2017*). We deem it therefore possible that the respective Mn$^{2+}$-driven activation of TORC1 is more specifically related to the architecture of its catalytic center. Resolution of this issue will therefore require future molecular dynamic modeling and/or structural analyses.

Our current study also highlights a need for coordinated control of TORC1 activity and cytoplasmic Mn$^{2+}$ levels. The latter critically depend on a set of conserved NRAMP transporters, such as yeast Pmr1 and Smf1/2, and their mammalian homologs ATP2C1/SPCA1 (*Muncanovic et al., 2019*) and SLC11A2 (DMT1; *Nevo and Nelson, 2006*; and this work). In yeast, cytoplasmic Mn levels can be increased by the loss of Pmr1, which causes a defect in sequestration of Mn$^{2+}$ in the Golgi. Alternatively, they can also be boosted by increased Smf1-mediated Mn$^{2+}$ uptake combined with Smf2-mediated Mn$^{2+}$ depletion in endosomes following the loss of Bsd2, an adaptor protein for the NEDD4 family E3 ubiquitin ligase Rsp5 (*Liu et al., 1997*) that normally favors vacuolar degradation of Smf1/2 (*Jensen et al., 2009*; *Portnoy et al., 2000*). In both cases, that is, loss of Pmr1 or of Bsd2 (specifically upon addition of Mn$^{2+}$ in the medium), enhanced cytoplasmic Mn$^{2+}$ levels mediate higher TORC1 activity and hence the resistance of cells to low doses of rapamycin. Conversely, our expression and localization studies of GFP-Smf1 and Smf2-GFP indicate that TORC1, while being controlled by Mn$^{2+}$ levels, also employs feedback control mechanisms that regulate Smf1 expression and Smf2 turnover. These observations are consistent with the recent paradigm shift assigning TORC1 an essential role in cellular and organismal homeostasis (*Eltschinger and Loewith, 2016*) and extend this role of TORC1 to Mn metabolism. A more detailed understanding of these regulatory circuits involving TORC1 and NRAMP transporters (and vice versa) will, however, require future analyses of the specific subcellular pools of Mn$^{2+}$ within different organelles using electron-nuclear double resonance spectroscopy (*McNaughton et al., 2010*) or inductively coupled plasma mass spectrometry (*Liu et al., 2019*). Of note, Mn$^{2+}$ also acts as a cofactor for glutamine synthetase (*Wedler and Denman, 1984*) that produces the TORC1-stimulating amino acid glutamine (*Jewell et al., 2015*). Hence, Mn$^{2+}$ homeostasis, amino acid metabolism, and TORC1 may be subjected to even more intricate, multilayered feedback regulatory circuits.

Our finding that Mn$^{2+}$ activates TORC1/mTORC1 may have important implications in different biological contexts. For instance, previous work has shown that chronic exposure to high levels of Mn$^{2+}$ causes autophagic dysfunction and hence accumulation of compromised mitochondria in mammalian astrocytes due to reduced nuclear localization of TFEB (transcription factor EB), a key transcription factor that coordinates the expression of genes involved in autophagy (*Zhang et al., 2020*). Because mTORC1 inhibits TFEB (*Settembre et al., 2012*) and mitochondrial quality can be improved by rapamycin-induced TFEB induction (and consequent stimulation of autophagy) (*Siddiqui et al., 2015*), our study now provides a simple rationale for the observed accumulation of damaged

mitochondria upon $Mn^{2+}$ exposure. Accordingly, excess $Mn^{2+}$ antagonizes autophagy and mitophagy at least in part through mTORC1 activation and subsequent TFEB inhibition, which prevents proper disposal of hazardous and reactive oxygen species-producing mitochondria. This model also matches well with our previous observation that rapamycin restricts cell death associated with anomalous mTORC1 hyperactivation (*Villar et al., 2017*). Finally, our data indicate that $Mn^{2+}$-driven TORC1 activation and the ensuing inhibition of auto- and mitophagy are also employed by yeast cells, which highlights the primordial nature of these processes.

Another area where our findings may be relevant relates to the use of $Mn^{2+}$ to stimulate anti-tumor immune responses. Recent studies have shown that $Mn^{2+}$ is indispensable for the host defense against cytosolic, viral double-stranded DNA as it mediates activation of the DNA sensor CGAS (cyclic GMP-AMP synthase) and its downstream adaptor protein STING1 (stimulator of interferon response cGAMP interactor 1) (*Wang et al., 2018*). Since $Mn^{2+}$ stimulates the innate immune sensing of tumors, Mn administration has been suggested to provide an antitumoral effect and improve the treatment of cancer patients (*Lv et al., 2020*). Nevertheless, mTORC1 hyperactivation is known to promote tumor progression (*Mossmann et al., 2018*), and carcinoma and melanoma formation have previously been associated with mutations in the human *PMR1* ortholog *ATP2C1* that cause Hailey-Hailey disease (*Mohr et al., 2011*). Based on our findings, it is therefore possible that tumor formation in these patients may be causally linked to $Mn^{2+}$-dependent DNA replication defects (*García-Rodríguez et al., 2012*), stress response (*Cialfi et al., 2016*), and mTORC1 activation.

In addition to the potential relevance of our findings in the context of chronic $Mn^{2+}$ exposure and immune stimulation, our findings may also provide new perspectives to our understanding of specific neuropathies. For instance, exposure to Mn dust or Mn containing smoke, as a byproduct of metal welding, is well known to cause a parkinsonian-like syndrome named manganism, a toxic condition of Mn poisoning with dyskinesia (*Racette et al., 2001*). Interestingly, while dyskinesia has been connected to L-dopamine-mediated activation of mTORC1 (*Santini et al., 2009*), our findings suggest that $Mn^{2+}$-driven mTORC1 hyperactivation may impair autophagy and thereby contribute to neurological diseases (*Cheng et al., 2020*). In line with this reasoning, compounds that inhibit mTORC1 activity, and thus stimulate autophagy, have been suggested as therapeutics for the treatment of Parkinson-like neurological symptoms (*Ravikumar et al., 2004*). In this context, Huntington disease (HD) is another example of patients suffering from a proteinopathy characterized by parkinsonian-like neurological syndromes (*McColgan and Tabrizi, 2018*). In HD patients, expansion of the polyglutamine tract in the N-terminus of the huntingtin protein leads to protein aggregation (*Macdonald, 1993*) and, intriguingly, HD patients exhibit reduced $Mn^{2+}$ levels in the brain (*Bowman et al., 2011*). This raises the question of whether the respective cells aim to evade Mn-driven mTORC1 activation, for example, by reducing Mn uptake or sequestration of $Mn^{2+}$ within organelles, to stimulate huntingtin protein degradation via autophagy. Finally, $Mn^{2+}$ also contributes to prion formation in yeast (*Chakrabortee et al., 2016*), and elevated $Mn^{2+}$ levels have been detected in the blood and the central nervous system of Creutzfeldt-Jakob patients (*Hesketh et al., 2008*). It will therefore be exciting to study the cell type-specific impact of $Mn^{2+}$-driven mTORC1 activation on metabolism, genome stability, checkpoint signaling, and the immune response, all processes that play a key role in neurological diseases and aging-related processes.

## Materials and methods
### Antibodies
Primary antibodies against GFP (Takara Bio, 632380, AB_10013427; 1:5000), G-6-PDH (Sigma-Aldrich, A9521, AB_258454; 1:5000), Adh1 (Millipore, 126745, AB_564196; 1:200000), RPS6 (Cell Signaling Technology, 2217, AB_331355; 1:1000), phospho-RPS6 (Ser$^{235-236}$) (Cell Signaling Technology, 4856, AB_2181037; 1:1000), RPS6KB (Cell Signaling Technology, 2708, AB_390722; 1:1000), phospho-RPS6KB (Thr$^{389}$) (Cell Signaling Technology, 9205, AB_330944; 1:1000), EIF4EBP1 (Cell Signaling Technology, 9452, AB_331692; 1:1000), phospho-EIF4EBP1 (Thr$^{37/46}$) (Cell Signaling Technology, 2855, AB_560835; 1:1000), phospho-EIF4EBP1 (Ser$^{65}$) (Cell Signaling Technology, 9451, AB_330947; 1:1000), AKT1 (Cell Signaling Technology, 4691, AB_915783; 1:1000), phospho-AKT(Ser$^{473}$) (Cell Signaling Technology, 4060, AB_2315049; 1:1000), phospho-AKT(Thr$^{308}$) (Cell Signaling Technology, 13038, AB_2629447; 1:1000), ULK1 (Cell Signaling Technology, 8359, AB_11178668; 1:1000), phospho-ULK1(Ser$^{757}$) (Cell

Signaling Technology, 6888, AB_10829226; 1:1000), MAP1LC3 AB (Cell Signaling Technology, 12741, AB_2617131; 1:1000), β-actin (Cell Signaling Technology, 4967, AB_330288; 1:1000) were used.

## Yeast strains, plasmids, and growth conditions

Yeast strains and plasmids used in this study are available upon request and listed in *Supplementary file 2*; *Supplementary file 3*, respectively. Gene disruption and tagging were performed with standard high-efficiency recombination methods. To generate strain MP6988, pSIVu-SMF1p-GFP-SMF1 was digested with *PacI* and transformed into *smf1Δ* strain YOL182C. Yeast cells were grown to mid-log phase in synthetic defined medium (SD; 0.17% yeast nitrogen base, 0.5% ammonium sulfate, 2% glucose, 0.2% drop-out mix) at 30°C. To induce autophagy, mitophagy, and Rtg1-Rtg3 retrograde signaling, cells were treated with 200 ng/ml rapamycin (Biosynth Carbosynth, AE27685) for the indicated time. For telomere length analyses, cells were grown for 3 days in rich medium (YPAD; 1% yeast extract, 2% peptone, 2% glucose, 0.004% adenine sulfate) with or without the addition of 10 mM $CaCl_2$. For imaging of Rtg3-GFP, cells were grown at 26°C in SD-MSG-ura medium (0.17% yeast nitrogen base, 0.5% monosodium glutamate, 2% glucose, 0.2% drop-out mix-ura) to exponential phase and treated or not with 200 ng/ml rapamycin for 30 min. For GFP1-Smf1 and Smf2-GFP analyses, cells were grown in SD-ura medium. Overnight precultures were quickly spun and diluted into SD-ura medium devoid of manganese sulfate (0.19% yeast nitrogen base without manganese sulfate [Formedium; CYN2001], 0.5% ammonium sulfate, 2% glucose, and 0.2% drop-out mix-ura). After 5–6 hr of growth, cells were treated with either rapamycin (200 ng/ml), CHX (25 μg/ml), or both compounds for the indicated times.

## Yeast cell lysate preparation and immunoblotting

Cells grown to mid-log phase were treated with 6.7% trichloroacetic acid (TCA, final concentration), pelleted, washed with 99% acetone, dried, dissolved in urea buffer (6 M urea, 50 mM Tris-HCl pH 7.5, 1% SDS, 1 mM PMSF, and 10 mM NaF) and disrupted with glass beads using a Yasui Kikai homogenizer. Cell lysates were heated at 65°C for 10 min in Laemmli SDS sample buffer, centrifuged at 15,000 *g* for 1 min, and the supernatants were subjected to SDS-PAGE and immunoblotted. Heat denaturation of samples was omitted to detect GFP-Smf1 and Smf2-GFP. Chemiluminescence signals were captured in an Amersham ImageQuant 800 Imager and quantified with ImageQuant TL software (Cytiva).

## Drug sensitivity assays

Yeast cells grown to mid-log phase were adjusted to an initial $A_{600}$ of 0.2, serially diluted 1:10, and spotted onto plates without or with rapamycin at the indicated concentrations (see figure legends). One mM $MnCl_2$ and/or 10 mM $CaCl_2$ were added to the medium when indicated. Plates were then incubated at 30°C for 3–4 days. Two or more biological replicates were performed for all conditions.

## Analysis of telomere length

Genomic DNA was isolated from yeast strains grown in YPAD for 3 days with or without the addition of 10 mM $CaCl_2$. DNA was digested with *XhoI*, separated on a 1% agarose-Tris borate EDTA gel, transferred to a Hybond XL (Amersham Biosciences) membrane, and hybridized with a $^{32}P$-labeled DNA probe specific for the terminal Y' telomere fragment. The probe was generated by random hexanucleotide-primed DNA synthesis using a short Y'-specific DNA template, which was generated by PCR from genomic yeast DNA using the primers Y up (5'-TGCCGTGCAACAAACACTAAATCAA-3') and Y' low (5'-CGCTCGAGAAAGTTGGAGTTTTTCA-3'). Two biological replicates of the whole experiment were conducted.

## In vitro TORC1 kinase assays

TORC1 was purified and kinase assays were performed as previously described (*Nicastro et al., 2021*) with minor modifications. For the kinase assays in the presence of various concentrations of magnesium or manganese, reactions were performed in a total volume of 20 μl with kinase buffer (50 mM HEPES/NaOH [pH 7.5], 150 mM NaCl), 400 ng of purified $His_6$-Lst4$^{Loop}$, 60 ng TORC1 (quantified based on the Tor1 subunit) and 640, 320, 160, 80, 40, or 20 μM $MgCl_2$/$MnCl_2$. The reactions were started by adding 1.4 μl of ATP mix (18 mM ATP, 3.3 mCi [γ-$^{32}P$]-ATP [Hartmann Analytic, SRP-501]). For the kinase

assays in the presence of various concentrations of ATP, reactions containing 4 mM $MgCl_2$, 160 µM $MnCl_2$, or both were started by adding 1.4 µl of stock ATP mix (5.7 mM ATP, 3.3 mCi [$\gamma$-$^{32}$P]-ATP [Hartmann Analytic, SRP-501]) or twofold serial dilutions. After the addition of sample buffer, proteins were separated by SDS-PAGE, stained with SYPRO Ruby (Sigma-Aldrich, S492) (loading control), and analyzed using a phosphoimager (Typhoon FLA 9500; GE Healthcare). Band intensities from three technical replicates were quantified with ImageJ and data were analyzed with GraphPad Prism using the Michaelis-Menten non-linear fitting.

### Cell culture

U2OS and HEK293T cell lines were obtained from the American Type Culture Collection (ATCC). GFP-MAP1LC3-expressing U2OS cells were kindly provided by Dr Eyal Gottlieb (Cancer Research UK, Glasgow, UK). All the cell lines were grown in high glucose DMEM base medium (Sigma-Aldrich, D6546) supplemented with 10% of fetal bovine serum, glutamine (2 mM), penicillin (Sigma-Aldrich, 100 µg/ml), and streptomycin (100 µg/ml), at 37°C, 5% $CO_2$ in a humidified atmosphere. Amino acid starvation was performed with EBSS medium (Sigma-Aldrich, E2888) supplemented with glucose at a final concentration of 4.5 g/l. When indicated, the starvation medium was complemented with $MnCl_2$ to a final concentration of 0.05–1 mM, chloroquine (Sigma-Aldrich; 10 µM), and amino acids, by adding MEM Amino Acids (Sigma-Aldrich, M5550), plus MEM Non-essential Amino Acid Solution (Sigma-Aldrich, M7145) and glutamine (2 mM). The identities of the cell lines bought from ATCC were authenticated by STR profiling. The identity of the GFP-MAP1LC3-expressing U2OS cells was verified by the donating research group. In addition, all cell lines were tested negative for mycoplasma contamination.

### Confocal and fluorescence microscopy

Yeast cells: For imaging of GFP-Tor1, GFP-Smf1, and Smf2-GFP, images were captured with an inverted spinning disk confocal microscope (Nikon Ti-E, VisiScope CSU-W1, Puchheim, Germany) that was equipped with an Evolve 512 EM-CDD camera (Photometrics), and a 100 × 1.3 NA oil immersion Nikon CFI series objective (Egg, Switzerland). Images were then processed using the FIJI-ImageJ software. For imaging of Rtg3-GFP, images were obtained by projection of a focal plane image derived from wide-field fluorescence microscopy (DM-6000B, Leica) at 100× magnification using L5 filters and a digital charge-coupled device camera (DFC350, Leica). Pictures were processed with LAS AF (Leica).

Human cells: $8 \times 10^5$ cells were grown in coverslips for 24 hr and treated for 4 hr as indicated. Thereafter, cells were fixed with 4% paraformaldehyde (Sigma-Aldrich) in PBS for 10 min at room temperature. For GFP-MAP1LC3 assessment, after three washes with PBS, coverslips were mounted with Prolong containing DAPI (Invitrogen). Samples were imaged with a Zeiss Apotome microscope. GFP aggregation in microscopy images was assessed using ImageJ software.

### Statistical analyses

Statistical analyses were performed using GraphPad Prism 7.0. Statistical significance was determined from at least three independent biological replicates using either Student's t-test or ANOVA followed by Tukey's multiple comparison test. Two-tailed Student's t-test was used for comparison of the means of two different experimental conditions. Paired t-test was used for the pairwise comparison of normalized data. ANOVA test was used for the analyses of three variables. Differences with a p-value lower than 0.05 were considered significant. *$p<0.05$; **$p<0.01$; ***$p<0.001$; ****$p<0.0001$. The number of independent experiments (n), specific statistical tests, and significance are described in the figure legends.

## Acknowledgements

We thank María Díaz de la Loza and Benjamin Pillet for scientific illustration work, and V Albanèse, E de Nadal, V Goder, KD Hirschi, C Ungermann, and E Gottlieb for plasmids, yeast strains, and cell lines. Research was funded by grants from the University of Seville (2020/00001326), Junta de Andalucía/ European Union Regional Funds (P20-RT-01220) and EMBO (STF-8685) to REW; the Swiss National Science Foundation (310030_166474/184671) to CDV; the Spanish Ministry of Science, Innovation and Universities (PGC2018-096244-B-I00) to RD. The author LZ was the recipient of a predoctoral grant from the Spanish Ministry of Science, Innovation and Universities (FPU19/04914).

## Additional information

### Funding

| Funder | Grant reference number | Author |
|---|---|---|
| Universidad de Sevilla | 2020/00001326 | Hélène Gaillard<br>Ralf Erik Wellinger |
| Junta de Andalucía | P20-RT-01220 | Ralf Erik Wellinger |
| European Molecular Biology Organization | STF-8685 | Ralf Erik Wellinger |
| Schweizerischer Nationalfonds zur Förderung der Wissenschaftlichen Forschung | 310030_166474/184671 | Claudio De Virgilio |
| Ministerio de Ciencia, Innovación y Universidades | PGC2018-096244-B-I00 | Raúl V Durán |
| Ministerio de Ciencia, Innovación y Universidades | FPU19/04914 | Laura Zarzuela |

The funders had no role in study design, data collection and interpretation, or the decision to submit the work for publication.

### Author contributions

Raffaele Nicastro, Conceptualization, Formal analysis, Investigation, Methodology, Writing – review and editing; Hélène Gaillard, Conceptualization, Formal analysis, Investigation, Visualization, Methodology, Writing – review and editing; Laura Zarzuela, Elisabet Fernández-García, Investigation; Marie-Pierre Péli-Gulli, Néstor García-Rodríguez, Conceptualization, Investigation, Methodology, Writing – review and editing; Mercedes Tomé, Conceptualization, Investigation, Methodology; Raúl V Durán, Conceptualization, Resources, Data curation, Formal analysis, Supervision, Funding acquisition, Methodology, Writing – review and editing; Claudio De Virgilio, Conceptualization, Resources, Data curation, Formal analysis, Supervision, Funding acquisition, Methodology, Writing – original draft, Writing – review and editing; Ralf Erik Wellinger, Conceptualization, Resources, Data curation, Formal analysis, Supervision, Funding acquisition, Investigation, Methodology, Writing – original draft, Project administration, Writing – review and editing

### Author ORCIDs

Raffaele Nicastro (ID) http://orcid.org/0000-0002-5420-2228
Hélène Gaillard (ID) http://orcid.org/0000-0002-5740-0641
Marie-Pierre Péli-Gulli (ID) http://orcid.org/0000-0002-6908-7082
Néstor García-Rodríguez (ID) http://orcid.org/0000-0002-4049-1604
Claudio De Virgilio (ID) http://orcid.org/0000-0001-8826-4323
Ralf Erik Wellinger (ID) http://orcid.org/0000-0002-4421-6618

### Decision letter and Author response

Decision letter https://doi.org/10.7554/eLife.80497.sa1
Author response https://doi.org/10.7554/eLife.80497.sa2

## Additional files

### Supplementary files

- Supplementary file 1. RTG1-3 target genes down-regulated in *pmr1Δ* cells.
- Supplementary file 2. Plasmids used in this study.
- Supplementary file 3. Yeast strains used in this study.
- MDAR checklist

## Data availability

All data generated or analyzed during this study are included in the manuscript and supporting files. Source data files have been provided for Figure 1, Figure 1-figure supplement 1, Figure 2, Figure 3, Figure 4 and Figure 5. The data set entitled 'Expression data of pmr1Δ mutants', originally published in García-Rodríguez et al (2012) and reused in this study is accessible at Gene Expression Omnibus under the accession code GSE29420.

The following previously published dataset was used:

| Author(s) | Year | Dataset title | Dataset URL | Database and Identifier |
| --- | --- | --- | --- | --- |
| Garcia-Rodriguez N, Wellinger RE | 2012 | Expression data of pmr1Δ mutants | https://www.ncbi.nlm.nih.gov/geo/query/acc.cgi?acc=GSE29420 | NCBI Gene Expression Omnibus, GSE29420 |

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
