## [Editor Report]

Your manuscript uses budding yeast to uncover a new input for the central metabolic regulator TOR complex 1 (TORC1) – manganese (Mn) levels and also demonstHelene rates that this dependence on Mn is conserved in humans. The combination of both in vivo and in vitro approaches as well as the demonstration of conservation of this phenomenon make the manuscript both broad and deep. TORC1 is already clearly a central coordinator of multiple inputs to guide cellular decisions of catabolism vs anabolism. Information on an additional way to modulate its activity is highly influential on both basic cell biology as well as therapeutic research.

---

## [Decision Letter]

**Decision letter after peer review:**

Thank you for submitting your article "Manganese is a Physiologically Relevant TORC1 Activator in Yeast and Mammals" for consideration by *eLife*. Your article has been reviewed by Maya Schuldiner as the Reviewing Editor and Jonathan Cooper as the Senior Editor.

Comments for the authors:

I have read the reviewers' comments from a previous submission and find that the authors did a very good job of answering the reviewers' concerns and have therefore decided not to send the manuscript out for another round of reviews. However, there are some small concerns of my own that I would like the authors to please pay attention to before the final submission.

1. Please add MW values on all westerns. This was requested by the previous reviewers and is still not properly done. While the authors answered "The entire blots including no GFP fusion controls and molecular weight markers are available upon request and may be provided as supplemental material if necessary" I do not think this is necessary at present but just lines representing the MW standards running height will suffice.

2. Figure 1E Please add MW markers (length in bp). I suggest also clearly stating on the image that it's a Southern blot of Genomic DNA.

3. Figure 1F – please write clearly in the Results section that SLC11A1 did not complement the phenotype.

4. Figure 2F should be performed in triplicate and quantified, please.

5. Figure 3A/B – please write on the gel what is being detected in each panel (Autoradiography (lower blot) and Sypro Ruby staining (upper blot).)

6. For Figure 4 it should be written much more clearly in the "Results" section that the authors used N' tagged Smf1 under its own promoter and from a non-endogenous locus and C' tagged Smf2 from its endogenous locus as well as reference that these are functional tags. At present, it requires some digging to figure this out.

7. Figure 4B the legend says that what is being quantified is "Quantification of GFP-Smf1. Percentages of GFP-Smf1 relative to the total GFP signal (GFP-Smf1 + GFP)" But the Y axis of the figure says only GFP-Smf1/2% – please clarify which is correct and write consistently.

8. In Figure 5F/G please write clearly that the third condition is -AA +Mn (right now the AA status is not written).

---

## [Author Response]

Essential revisions:I have read the reviewers' comments from a previous submission and find that the authors did a very good job of answering the reviewers' concerns and have therefore decided not to send the manuscript out for another round of reviews. However, there are some small concerns of my own that I would like the authors to please pay attention to before the final submission.1. Please add MW values on all westerns. This was requested by the previous reviewers and is still not properly done. While the authors answered "The entire blots including no GFP fusion controls and molecular weight markers are available upon request and may be provided as supplemental material if necessary" I do not think this is necessary at present but just lines representing the MW standards running height will suffice.

We have added the running height of MW standards on all immunoblots as requested.

2. Figure 1E Please add MW markers (length in bp). I suggest also clearly stating on the image that it's a Southern blot of Genomic DNA.

MW marker sizes and ‘Southern blot of Genomic DNA’ have been added to Figure 1E.

3. Figure 1F – please write clearly in the Results section that SLC11A1 did not complement the phenotype.

This is now clearly stated in the results, which now reads: ‘Expression of the SLC11A1 did not rescue the lack of Smf2, leading to poor growth even in the absence of rapamycin. However, the SLC11A2 isoform complemented Smf2 function in these assays (Figure 1F), indicating that Pmr1 and Smf2 are evolutionarily conserved transporters that are required for Mn^2+^ homeostasis.’

4. Figure 2F should be performed in triplicate and quantified, please.

The experiment shown in Figure 2F was performed four times and we have now added the quantification of Rtg3-GFP signal in Figure 2G. Results, Figure Legend, and Figure2_Source data 1 have been appropriately modified.

5. Figure 3A/B – please write on the gel what is being detected in each panel (Autoradiography (lower blot) and Sypro Ruby staining (upper blot).)

We have now labeled each blot.

6. For Figure 4 it should be written much more clearly in the "Results" section that the authors used N' tagged Smf1 under its own promoter and from a non-endogenous locus and C' tagged Smf2 from its endogenous locus as well as reference that these are functional tags. At present, it requires some digging to figure this out.

We have now clarified this in the Results section, which now reads: ‘…we next asked whether rapamycin-mediated TORC1 inactivation may affect the levels and/or localization of Smf1 and Smf2 using a strain that expresses N-terminally tagged Smf1 (GFP-Smf1) under its own promoter from a non-endogenous locus or a strain that expresses C-terminally tagged Smf2 (Smf2-GFP) from its endogenous locus (García-Rodríguez et al., 2015; Renz et al., 2020).’

7. Figure 4B the legend says that what is being quantified is "Quantification of GFP-Smf1. Percentages of GFP-Smf1 relative to the total GFP signal (GFP-Smf1 + GFP)" But the Y axis of the figure says only GFP-Smf1/2% – please clarify which is correct and write consistently.

We apologize for the confusing labeling, we have now changed the Y axis label which now reads ‘GFP-Smf1/Total [%]’ and ‘Smf2-GFP/Total [%]’, respectively.

8. In Figure 5F/G please write clearly that the third condition is -AA +Mn (right now the AA status is not written).

We have changed the labeling of Figures 5F and 5G, which now clearly indicates whether the cells were starved or not in each condition.

[Editors' note: we include below the reviews that the authors received from another journal, along with the authors’ responses.]

Reviewer #1:Nicastro et al. do a very good job of bringing clarity to prior observations linking Mn^2+^ to TORC1 signaling. This new understanding is important for the TOR field and will have much broader implications to those studying Mn^2+^ dysregulation and associated pathologies. Topic-wise, I feel this work is well suited for publication in PNAS. However, I would recommend several concerns be addressed prior to acceptance.Major comments1. "Rapamycin induced the degradation of GFP-Smf1 during a 2h-treatment (as measured by the increase of GFP signals on immunoblots) in WT, pmr1∆, and bsd2∆ cells (Figure 2A and 2B)" – this is not particularly evident from the data in this figure, particularly for WT cells where rapamycin increase GFP-Smf1 levels at shorter time points. The data would be more consistent with reduced expression (rapamycin is well known to block translation).

We have generated a new strain in which GFP-Smf1 is integrated into the genome under the control of its endogenous promoter (instead of using a plasmid-borne GFP-Smf1 reporter) and repeated these experiments with wild type cells grown on media containing low manganese levels to enhance the starting levels of Smf1-GFP at the plasma membrane. Our new data show that GFP-Smf1 and cleaved GFP signals increase upon treatment with rapamycin (new Figure 4). To assess whether the rapamycin-induced increase in GFP-Smf1 is due to reduced protein turnover, or occurs at transcriptional and/or translational level, we have used cycloheximide (with and without rapamycin) treatments in parallel experiments. Our new results indicate that GFP-Smf1 expression is increased at the transcriptional level upon rapamycin treatment, which is also in line with published literature. We conducted the same set of experiments with Smf2-GFP-expressing cells. Here, our experiments indicate that rapamycin treatment enhances the overall turnover of Smf2. As suggested below (comment number 4), we have removed the experiments with *pmr1∆* and *bsd2∆* cells, which were difficult to interpret.

2. "fluorescence microscopy revealed that, in exponentially growing WT, pmr1∆, and bsd2∆ cells, GFP-Smf1 localized weakly, moderately, and strongly, respectively, to the plasma membrane, while rapamycin treatment induced the degradation of GFP-Smf1 as visualized by the appearance of stronger GFP signals in the vacuoles in all three mutants Figure 2D" – I am not sure this would stand up to proper quantitation of these microscopy data. "Thus, TORC1 promotes the expression of GFP-Smf1 at the plasma membrane and inhibits its lysosomal degradation." – this reviewer is unconvinced by the data provided. Is such regulation even germane to the main conclusions of this manuscript?

We agree with this reviewer that the issue of regulation is not germane to the main conclusions of the paper. We have repeated the microscopy data in cells expressing either GFP-Smf1 or Smf2-GFP together with a vacuolar marker protein (Vph1-mCherry; see also answer to point 1; new Figure 4).

3. Rapamycin-induced turnover of Smf2, at least in WT cells appears to be better supported by the data. Again, better quantitation of the microscopy images is warranted.

As described in our response to point 1, we have repeated the immunoblot analyses in wild-type cells treated either with rapamycin, cycloheximide, or rapamycin and cycloheximide combined to assess the rapamycin-induced turnover of Smf2 (new Figure 4). New microscopy data showing the localization of Smf2-GFP and the vacuolar marker Vph1-mCherry have been generated and analyzed.

4. Globally, these experiments are complicated by the fact that rapamycin efficacy as a TORC1 inhibitor is inherently different in these strains. This reviewer's suggestion would be to focus only on Smf1 and Smf2 turnover in WT cells treated with rapamycin – concluding that Smf1 regulation is not readily apparent whereas Smf2 regulation is.

We agree with this comment and have focussed on Smf1 and Smf2 turnover in wild-type cells as outlined under point 1 above.

5. Figure 3A-D and associated text. The relevant comparison is between the green and purple columns (i.e. to argue that smf2D suppresses the pmr1D phenotype). Why is this comparison not made?

We have extended the quantification of the immunoblots and now show a comparison of the total amount of GFP signal between the strains 6 h after rapamycin addition and the percentage of cleaved GFP for each strain at this same timepoint (new Figure 2, B and D). The comparison of *pmr1∆* and *pmr1∆ smf2∆* is now appropriately described.

6. "Mn has been shown to activate the mTORC1 upstream kinase AKT1 (AKT serine/threonine kinase 1) (53) and we find that MnCl_2_ also induced an increase in AKT1 phosphorylation in amino-acid starved cells. However, AKT1 phosphorylation appeared to be much less pronounced than the phosphorylation of the mTORC1 downstream targets, thus validating the specificity of Mn towards mTORC1." I think the authors are too quick to dismiss the possibility that mTORC2 is also activated by Mn. Why not leave open this possibility?

We have revised the text to leave the possibility open that mTORC2 activity may also be regulated by Mn^2+^. The text now reads: ‘In line with our finding that MnCl_2_ retained AKT1 (AKT serine/threonine kinase 1) phosphorylation in amino-acid starved cells, Mn^2+^ has been reported to activate the mTORC1 upstream kinase AKT1, which is also a known mTORC2 downstream target (55, 56). However, AKT1 phosphorylation appeared to be much less pronounced than the phosphorylation of the mTORC1 downstream targets, validating the specificity of Mn towards mTORC1. Whether Mn^2+^ may stimulate TORC2 activity remains to be explored.’

7. What is the mechanism by which Mn^2+^ renders TORC1 activity insensitive to rapamycin? I don't understand how elevated Mn^2+^ can negate the effects of 200 nM rapamycin. To what rapamycin concentration are pmr1 cells resistant? Are these cells also resistant to caffeine, wortmannin, or ATP-competitive TOR inhibitors? Does Mn^2+^ make TORC1 resistant to FKBP12-rapamycin in vitro?

Please note that we do not claim that Mn^2+^ renders TORC1 insensitive to rapamycin. Our results show that elevated cytoplasmic Mn^2+^ levels render *pmr1∆* cells resistant to low levels of rapamycin (10 ng/ml). The rapamycin resistance of *pmr1∆* cells is already less pronounced when applying slightly higher rapamycin concentrations of 20 ng/ml (data available upon request). Hence, Mn^2+^ simply enhances the resistance of cells to rapamycin. Importantly, however, as described in the answer to point 8, we have extended our in vitro analyses, which now include both kinetic experiments with different ATP concentrations and competition experiments in the presence of both Mn^2+^ and Mg^2+^ ions. Our new data support a mechanism according to which Mn^2+^ leads to a more efficient ATP coordination in the catalytic cleft of TORC1 than Mg^2+^. Hence, Mn^2+^ boosts the activity of TORC1, which increases the threshold level at which rapamycin can efficiently inhibit TORC1. The question of whether *pmr1∆* cells are also resistant to other TORC1 inhibitors is interesting, but many of these inhibitors (such as caffeine and wortmannin) are not specific for TORC1 and thus seem not appropriate for this purpose. In addition, the ATP-competitive TOR inhibitor torin 1 is quite specific in human and *S. pombe* cells, but has not been reported to function in *S.cerevisiae*, likely because of the poor permeability/uptake in budding yeast cells. Finally, in vitro experiments including purified FKBP12-rapamycin complexes are, in our view, unlikely to provide more mechanistic insight.

8. Does Mg^2+^ compete with Mn^2+^ in in vitro kinase assays? I.e. are both acting as co-factors for the ATP hydrolysis or does Mn^2+^ potentially have an allosteric mode of activation as indicated in the cartoon model / discussion.

This is a very interesting issue. Because TORC1 is inactive without Mg^2+^ or Mn^2+^, but active with either one of these ions, both must play a role in the catalytic center of the kinase. To address the point raised here further, we have measured the Km for ATP of TORC1 with Mn^2+^ and with Mg^2+^ (new Figure 3). Furthermore, we have performed competition experiments in the presence of both ions, uncovering that Mn^2+^ activates TORC1 substantially better than Mg^2+^, which is primarily due to its capacity to lower the K_m_ for ATP and hence mediating more efficient ATP coordination in the catalytic cleft of TORC1.

Minor comments:9. Abstract: "activation of TORC1 cause retrograde dysregulation" – what is retrograde dysregulation? The authors need to be more specific – there are multiple retrograde events in the cell.

We have reformulated the abstract, which is now more specific and the corresponding sentence reads: ‘genetic interventions that increase cytoplasmic Mn^2+^ levels antagonize the effects of rapamycin in triggering autophagy, mitophagy, and Rtg1-Rtg3-dependent mitochondrion-to-nucleus retrograde signaling.’

10. Introduction "Whether TORC1/mTORC1 is also able to sample the presence of trace elements is currently not known." This statement is not entirely correct: 10.1016/j.cmet.2012.10.001

We apologize for this mistake and have now included this reference in the introduction and modified the text accordingly, which now reads: ‘Interestingly, mTORC1 regulates cellular Fe homeostasis (30), but how it may be able to sense Fe levels remains largely unknown. In addition, although TORC1/mTORC1 requires divalent metal ions to coordinate ATP at its catalytic cleft (31, 32), it is currently not known whether these or any other trace elements may play a physiological or regulatory role in controlling its activity.’

11.Introduction "reveal that TORC1 protein activity is strongly activated" – protein kinase activity?

The text has been corrected.

12. Introduction "retrograde response activation" – again, should specify your meaning of retrograde

The text now specifies ‘Rtg1-3 transcription factor complex-dependent retrograde response activation’.

13. "data" is plural of datum. "This data" should be "these data" or "this dataset" etc.

Has been corrected in the revised manuscript.

14. Discussion "Nevertheless, MTOR (mechanistic target of rapamycin kinase)" a little late to be defining mTOR

Has been corrected.

15. vernacular usage of "This begs the question" – consider rephrasing

The text has been rephrased.

16. Throughout the manuscript the authors refer to Mn – metallic manganese. I can understand they do not want to commit to a specific oxidation state (although this could also be interesting / tested), but I am not sure that the Mn nomenclature is a better solution. Perhaps Mnx+?

We agree that using Mn was not the better solution. Since the physiological oxidation state of Mn is 2^+^ and accordingly, MnCl_2_ was used in our study in the experiments in which manganese was externally supplied, we have now used the divalent Mn^2+^ nomenclature throughout the manuscript except when we talk about manganese where we have used Mn.

Reviewer #2:These studies explore a role for high levels of Mn in controlling TOR activity. The findings nicely confirm previously published studies that indicate a role for elevated Mn in activating TOR in yeast cells (2007) and activation of mammalian TOR enzymatic activity in vitro with Mn (1999). In addition, the authors report that the reverse is true and TOR will regulate Nramp Mn transporter by inducing the turnover of these proteins in yeast cells. This indeed would represent a novel finding. However, none of the studies address Nramp protein stability and turnover, and so this conclusion appears premature. The studies showing Mn regulation of the TOR pathway in mammalian cells seem solid and these findings could be of general interest.Major comments:1. The title states "Mn is a physiologically relevant TORC1 activator". The authors should consider removing the word physiologically in the title and other places in the paper because all studies are conducted under non-physiological conditions of high or nearly toxic levels of Mn.

We would like to point out that most of our yeast studies were carried out with cells growing exponentially on normal media (without MnCl_2_ addition). The ones performed with nonphysiological MnCl_2_ levels are limited to Figure 1. Hence, we think that the title remains justified based on the full set of data presented.

2. The authors cannot claim that TORC1 regulates NRAMP transporter stability and turnover without conducting protein turnover studies (e.g., pulse chase or cycloheximide) to monitor SMF1 and SMF2 half lives. The authors themselves admit that the change in SMF1 levels in the pmr1 mutant may be due to a transcriptional, not protein degradation effects (bottom of page 6).

We have addressed this issue as outlined under point 1 of reviewer **#**1 and used cycloheximide and rapamycin and cycloheximide combined to assess Smf1 and Smf2 turnover.

3. There are numerous issues noted with the GFP fusion protein studies in Figures 2 and 3.a. Mutants of bsd2 have been published to express extremely high levels of SMF1 due to loss of SMF1 turnover. And yet in the same growth media, these studies show very little if any change in GFP-SMF1 in bsd2 strains (no RAP) in 2A and 2E. Thus one wonders whether GFP-SMF1 a valid tool to study SMF1 turnover. There was no information provided on how this was constructed. Is GFP at the N-terminus or C-terminus, and if at the N-terminus, wouldn't this impact trafficking?

Smf1 is N-terminally tagged and has previously been shown to be functional (Renz et al., J. Cell Science, 2020). In the revised manuscript, we have used a strain expressing GFP-Smf1 from its endogenous promoter at the *URA3* locus (instead of the plasmid-borne reporter used in the previous version of our manuscript). In addition, as suggested by reviewer #1 (point 4), we now focus the turnover studies on Smf1 and Smf2 in wild-type cells.

b. There are no molecular weight markers on any gels in this manuscript and the top portion of 2A and 2E gel is cut off so one cannot examine if there are other glycosylated species of SMF1 and SMF2. The no GFP fusion controls are missing in all studies. This is particularly important for 2A and 2E where the band for full length protein is very faint.

We have considerably improved the quality of the GFP-Smf1 and Smf2-GFP blots (now shown in Figure 4A and 4D). The entire blots including no GFP fusion controls and molecular weight markers are available upon request and may be provided as supplemental material if necessary.

c. The authors are using the ratio of free GFP to full length GFP-protein as indicators of protein turnover. Then why is the increase in free GFP not accompanied by a loss in full length protein (as example see pmr1 and bsd2 mutants in Figure 2A and 2D)? More direct methods and procedures to study protein turnover should be used (see also comment 2 above).

We have addressed this issue as outlined under point 1 of reviewer #1 and point 2 (this reviewer). The quantifications now show the percentage of uncleaved protein (*e.g.* GFP-Smf1) as compared to the total GFP signal (GFP-Smf1 + cleaved GFP), which reflects protein turnover (Arines and Li, STAR Protocols, 2022).

d. Mutants of pmr1 show low expression levels of all the GFP fusions of Figure 2 and Figure 3, even in the absence of RAP. This makes the results difficult to reconcile, particularly in 3E.

We have eliminated these GFP-fusion data concerning the pmr1∆ strain as they are not relevant for the main conclusions of this study (as discussed above; please see points 2 and 4 of reviewer #1). Regarding Rtg3-GFP (Figure 3E): inhibition of TORC1 causes both an increase in Rtg3GFP levels and cytoplasmic-to-nuclear transfer of Rtg3-GFP (as reported in Ruiz-Roig et al., Mol Biol Cell, 2012). Because loss of Pmr1 activates TORC1 and hence antagonizes rapamycin treatment, the respective cells have lower Rtg3-GFP levels. This in itself further supports our model and is in line with the conclusion that the *pmr1∆* cells are compromised for RTG signaling.

e. The microscopy images of Figures2 and 3 are difficult to interpret because it is not clear what is free GFP versus intact protein. Additionally, they authors the making conclusions regarding vacuolar and nuclear localization without markers of these organelles.

GFP is typically freed from fusion proteins once they reach the vacuolar lumen. To demarcate the vacuole, we have now used a fluorescently labeled vacuolar membrane protein (Vph1mCherry) alongside GFP-Smf1 and Smf2-GFP (new Figure 4). Regarding the former Figure 3 (now Figure 2), we did not use any nuclear marker as it is known that rapamycin induces cytoplasmic-to-nuclear transfer of Rtg3-GFP (Ruiz-Roig et al., Mol Biol Cell, 2012).

Minor comments:4. I would remove the word cofactor in the conclusion statement "Our study identifies Mn as a metal cofactor.." The authors have not demonstrated that TOR binds Mn in cells in vivo.

We have rephrased the sentence, which now reads: ‘Our study identifies Mn^2+^ as a divalent metal cofactor that stimulates the enzymatic activity of the TORC1 complex in vitro and suggests that Mn^2+^ functions similarly in vivo both in yeast and mammalian cells.*’*

5. I believe the legend for Figure 4A has the panels mixed up.

Thanks for pointing this out. This has been corrected.

6. Second sentence of second paragraph, I would change divalent metal transporter to Mn homeostasis proteins because not all these proteins are documented metal transporters. I would remove Mtm1, as this is now known to be a protein for mitochondrial Fe homeostasis, not Mn.

With the exception of Mtm1, for which controversial reports have been published, all listed proteins are documented metal transporters. To account for the lack of evidence about Mtm1, we have rephrased the text as follows: ‘several metal transporters have been associated with Mn^2+^ transport in yeast, including those localized at the plasma membrane (Smf1 and Pho84), the endosomes (Smf2 and Atx2), the Golgi (Pmr1 and Gdt1), the vacuole (Ccc1 and Vcx1), the ER (Spf1), and possibly the mitochondria (Mtm1).’ Note that Mtm1 is drawn with a question mark in the corresponding scheme (Figure 1A).

7. Second paragraph page 6: please define SLC11A1 and SLC11A2 as NRAMP1 and DMT1.

Is now defined in the revised manuscript.